# A Diachronic Perspective on User Trust in AI under Uncertainty

**Shehzaad Dhuliawala**[*]   **Vilém Zouhar**[*]   **Mennatallah El-Assady**   **Mrinmaya Sachan**

Department of Computer Science, ETH Zürich
{sdhuliawala,vzouhar,msachan}@inf.ethz.ch   menna.elassady@ai.ethz.ch

## Abstract

In a human-AI collaboration, users build a mental model of the AI system based on its reliability and how it presents its decision, e.g. its presentation of system confidence and an explanation of the output. Modern NLP systems are often uncalibrated, resulting in confidently incorrect predictions that undermine user trust. In order to build trustworthy AI, we must understand how user trust is developed and how it can be regained after potential trust-eroding events. We study the evolution of user trust in response to these trust-eroding events using a betting game. We find that even a few incorrect instances with inaccurate confidence estimates damage user trust and performance, with very slow recovery. We also show that this degradation in trust reduces the success of human-AI collaboration and that different types of miscalibration—unconfidently correct and confidently incorrect—have different negative effects on user trust. Our findings highlight the importance of calibration in user-facing AI applications and shed light on what aspects help users decide whether to trust the AI system.

## 1 Introduction

AI systems are increasingly being touted for use in high-stakes decision-making. For example, a doctor might use an AI system for cancer detection from lymph node images (Bejnordi et al., 2017), a teacher may be assisted by an AI system when teaching students (Cardona et al., 2023), or individuals may rely on AI systems to fulfill their information requirements (Mitra et al., 2018). AI systems are integrated across diverse domains, with an expanding presence in user-centric applications. Despite their growing performance, today's AI systems are still sometimes inaccurate, reinforcing the need for human involvement and oversight.

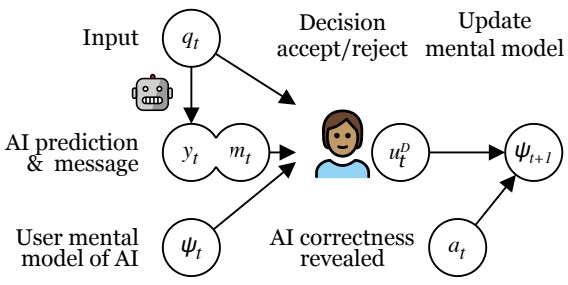

Figure 1: Diachronic view of a typical human-AI collaborative setting. At each timestep $t$, the user uses their prior mental model $\psi_t$ to accept or reject the AI system's answer $y_t$, supported by an additional message $m_t$ (AI's confidence), and updates their mental model of the AI system to $\psi_{t+1}$. If the message is rejected, the user invokes a fallback process to get a different answer.

An effective approach for facilitating decision-making in collaborative settings is for the AI system to offer its confidence alongside its predictions. This is shown in Figure 1, where the AI system provides an additional message that enables the user to either accept or reject the system's answer based on the additional message, such as the confidence score. This makes a strong case for the AI's confidence being calibrated (Guo et al., 2017) – when the confidence score aligns with the probability of the prediction being correct.

When a user interacts with an AI system, they develop a mental model (Hartson and Pyla, 2012) of how the system's confidence relates to the integrity of its prediction. The issue of trust has been extensively studied in psychology and cognitive science with Mayo (2015); Stanton et al. (2021) finding that incongruence (mismatch between mental model and user experience) creates distrust. Given the ever-increasing reliance on AI systems, it is crucial that users possess a well-defined mental model that guides their trust in these systems. Nevertheless, our current understanding regarding the evolution of user trust over time, its vulnerability to trust-depleting incidents, and the methods to re-

---

[*]Shared first authorship
[0]Data & code: github.com/zouharvi/trust-intervention

store trust following such events remain unclear. Addressing these inquiries holds great significance in the advancement of reliable AI systems.

In this paper, our objective is to investigate user interactions with an AI system, with a specific focus on how the system's confidence impacts these interactions. Through a series of carefully designed user studies, we explore the implications of miscalibrated confidences on user's perception of the system and how this, in turn, influences their trust in the system. Our experiments shed light on how users respond to various types of miscalibrations. We find that users are especially sensitive to confidently incorrect miscalibration (Section 4.1) and that the trust does not recover even after a long sequence of calibrated examples. Subsequently, we delve into an analysis of how trust degradation corresponds to the extent of miscalibration in the examples provided (Section 4.2). Then, we assess whether diminished trust in an AI system for a specific task can extend to affect a user's trust in other tasks (Section 4.3). We also explore different methodologies for modeling a user's trust in an AI system (Section 5). Our results show how reduced trust can lower the performance of the human-AI team thus highlighting the importance of holistic and user-centric calibration of AI systems when they are deployed in high-stakes settings.

## 2 Related Work

**Human-AI Collaboration.** Optimizing for cooperation with humans is more productive than focusing solely on model performance (Bansal et al., 2021a). Human-AI collaboration research has focused on AI systems explaining their predictions (Ribeiro et al., 2016) or examining the relationship between trust and AI system's accuracy (Rechkemmer and Yin, 2022; Ma et al., 2023). Related to our work, Papenmeier et al. (2019); Bansal et al. (2021b); Wang and Yin (2022); Papenmeier et al. (2022) examined the influence of explanations and found that inaccurate ones act as deceptive experiences which erode trust.

Nourani et al. (2021); Mozannar et al. (2022) study the development of mental models which create further collaboration expectations. This mental model, or the associated expectations, can be violated, which results in degraded trust in the system and hindered collaboration (Grimes et al., 2021). The field of NLP offers several applications where trust plays a vital role, such as chatbots for various tasks or multi-domain question answering (Law et al., 2021; Vikander, 2023; Chiesurin et al., 2023) and transparency and controllability are one of the key components that increase users' trust Bansal et al. (2019); Guo et al. (2022).

**Trust and Confidence Calibration.** A common method AI systems use to convey their uncertainty to the user is by its confidence (Benz and Rodriguez, 2023; Liu et al., 2023). For the system's confidence to reflect the probability of the system being correct, the confidence needs to be calibrated, which is a long-standing task (Guo et al., 2017; Dhuliawala et al., 2022). This can be any metric, such as quality estimation (Specia et al., 2010; Zouhar et al., 2021) that makes it easier for the user to decide on the AI system's correctness. Related to calibration is selective prediction where the model can abstain from predicting. The latter has been studied in the context of machine learning (Chow, 1957; El-Yaniv et al., 2010) and its various applications (Rodriguez et al., 2019; Kamath et al., 2020; Zouhar et al., 2023).

Trust calibration is the relation between the user's trust in the system and the system's abilities (Lee and Moray, 1994; Turner et al., 2022; Zhang et al., 2020; Yin et al., 2019; Rechkemmer and Yin, 2022; Gonzalez et al., 2020; Vodrahalli et al., 2022). Specifically, Vodrahalli et al. (2022) explore jointly optimization of calibration (transformation of the AI system reported confidence) with human feedback. They conclude that uncalibrated models improve human-AI collaboration. However, apart from their experimental design being different from ours, they also admit to not studying the temporal effect of miscalibrations. Because of this, our results are not in contradiction.

**Modeling User Trust.** Ajenaghughrure et al. (2019); Zhou et al. (2019) predictively model the user trust in the AI system. While successful, they use psychological signals, such as EEG or GSR, for their predictions, which is usually inaccessible in the traditional desktop interface setting. Li et al. (2023) use combination of demographic information together with interaction history to predict whether the user is going to accept or reject AI system's suggestion. The field has otherwise focused on theoretical frameworks to explain factors that affect trust in mostly human-robot interaction scenarios (Nordheim et al., 2019; Khavas et al., 2020; Ajenaghughrure et al., 2021; Gebru et al., 2022).

## 3 Human AI Interaction over Time

We begin by providing a preliminary formalism for a human-AI interaction over time. It comprises of two interlocutors, an **AI system** and a **user**. At time $t$, the user provides the AI system with an input or a question $q_t$ and the AI system responds with an answer $y_t$ along with a message comprising of its confidence in the answer $m_t$. The user has two options, either they accept the AI's answer or reject it and try to find an answer themselves. The AI is either **correct** ($a_t = 1$) or **incorrect** ($a_t = 0$). The combination of correctness $a_t$ and confidence $m_t$ results in four different possibilities each with a different reward, or risk, shown in Figure 2. For example, confidently incorrect may lead to the user disastrously accepting a false answer while unconfidently correct will make the user spend more time finding the answer themselves.

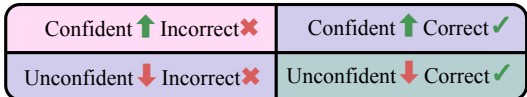

Figure 2: Possible correctness and confidence combinations of an AI system. Confidently incorrect and unconfidently correct are *miscalibrated* while the rest is *calibrated* (i.e. confidence corresponds to correctness.

During the interaction, the user learns a **mental model** ($\Psi_t$) of the AI system that they can use to reject and accept the AI's prediction. This mental model encapsulates something commonly referred to as **user trust**, which is, however, abstract and can not be measured directly. Instead, in our study, we rely on a proxy that describes a manifestation of this trust. We ask the user to make an estimate of their trust by tying it to a monetary reward. We assume that both depend on the given question $q_t$, message $m_t$, and history. The users place a bet between 0¢ and 10¢, i.e. $u_t^B = U^B(q_t, m_t, \Psi_t) \in [0¢, 10¢]$. We formally define the user's decision to accept or reject the AI's answer as $u_t^D = U^D(q_t, m_t, \Psi_t) \in \{1, 0\}$, given question $q_t$, message $m_t$, and history. In this work, by the user's mental model, we refer to it in the context of the features the user might use to decide how much they are willing to bet on the AI's prediction and how likely they are to agree with the AI and how $\Psi_t$ changes over time.

### 3.1 Study Setup

To study how user trust changes temporally we design a set of experiments with a sequence of interactions between a user and a simulated AI question-answering (QA) system. We recruit participants who are told that they will evaluate a QA system's performance on a sequence of question-answer pairs. The participants are shown the AI's produced confidence in its answer and then are instructed to use this confidence to assess its veracity. We term an instance of the AI's question, prediction, and confidence as a stimulus to the user. This method of using user interactions with a system to study user trust is similar to the study performed by Gonzalez et al. (2020). After the participant decides if the system is correct or incorrect, they bet from 0¢ to 10¢ on their decision about the system's correctness. We then reveal if the AI was correct or incorrect and show the user the gains or losses. The monetary risk is chosen intentionally in order for the participants to think deeply about the task. An alternative, used by Vodrahalli et al. (2022), is to simply ask for participants' confidence in the answer. While straightforward, we consider this to be inadequate in the crowdfunding setting. This decision is further supported by the fact that there is a difference between what participants report and what they do (Papenmeier et al., 2019). The average duration of the experiment was 6.7 minutes (Figure 9) and we collected 18k stimuli interactions (Table 3). See Figure 3 for an overview of the experiment design and Figure 13 for the annotation interface.[1]

### 3.2 Simulating AI

To investigate users' interactions, we simulate an AI system that outputs predictions and confidences. The prediction and confidence are produced using a pre-defined generative process.

Our simulated AI encompasses four modes for the generation of AI 'correctness' and confidence values. For miscalibrated questions, we have two modes: confidently incorrect (CI) and unconfidently correct (UC) modes, while for calibrated questions we use the accurate mode (control) to generate questions.

We define a conditional variable $c_t$ which denotes the aforementioned conditions. Then, based on the condition $c_t$, we have the following data generation process at timestep $t$. In our data generation process, we first decide the AI correctness $a_t \in [0, 1]$ and then decide the confidence $m_t \in [0\%, 100\%]$ as below:

---

[1]Online demo: zouharvi.github.io/trust-intervention

$$a_t \sim \begin{cases} \text{Bernoulli}(0.7) & \text{if } c_t = \text{calibrated} \\ \text{Bernoulli}(0.0) & \text{if } c_t = \text{CI} \\ \text{Bernoulli}(1.0) & \text{if } c_t = \text{UC} \end{cases}$$

$$m_t \sim \begin{cases} \text{Uniform}(0.45, 0.85) & \text{if } c_t = \text{cal.} \wedge a_t = 1 \\ \text{Uniform}(0.2, 0.55) & \text{if } c_t = \text{cal.} \wedge a_t = 0 \\ \text{Uniform}(0.7, 1.0) & \text{if } c_t = \text{CI} \ \wedge a_t = 0 \\ \text{Uniform}(0.1, 0.4) & \text{if } c_t = \text{UC} \wedge a_t = 1 \end{cases}$$

To control for participants prior knowledge of the answers to the provided questions, we use randomly generated questions with fictional premises. We also experimented with questions sourced from a combination of Natural Questions (Kwiatkowski et al., 2019) and TriviaQA (Joshi et al., 2017). Unfortunately, this approach resulted in a lot of noise and instances of misconduct as participants would look up the answers to increase their monetary reward. See Appendix A for a description of stimuli generation. We note that the set of questions that the participants see have similar ECE (Expected Calibration Error) scores and we compare this to a real NLP model in Appendix B.

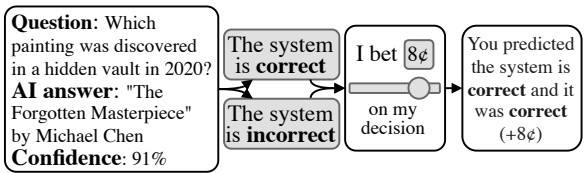

**Question**: Which painting was discovered in a hidden vault in 2020? **AI answer**: "The Forgotten Masterpiece" by Michael Chen **Confidence**: 91% → The system is **correct** / The system is **incorrect** → I bet 8¢ on my decision → You predicted the system is **correct** and it was **correct** (+8¢)

Figure 3: Pipeline for a single stimulus out of 60. The maximum payout for a bet is 10¢. UI Elements show possible user actions. See Figure 13 for screenshots.

## 4 Experiments

We perform three types of experiments. In Section 4.1, we establish the different effects of confidently incorrect and unconfidently correct stimuli. Then, in Section 4.2 we see how the size of confidently incorrect intervention affects the users interaction with the AI system and in Section 4.3 explore if miscalibration is transferable between question types. Lastly, we predict the user interaction in Section 5.

### 4.1 Effect of Miscalibration

We categorize AI behavior into four categories (Figure 2) and design an experiment to answer:

**RQ1:** Do miscalibrated examples affect user trust and alter how they interact with the AI system?

We posit that miscalibrated stimuli decrease user trust and subsequently verify the hypotheses:

**H1:** Confidently incorrect examples lower participants' trust in the system
**H2:** Unconfidently correct examples lower participants' trust in the system, but less so
**H3:** Miscalibrated examples reduce the human-AI collaboration performance

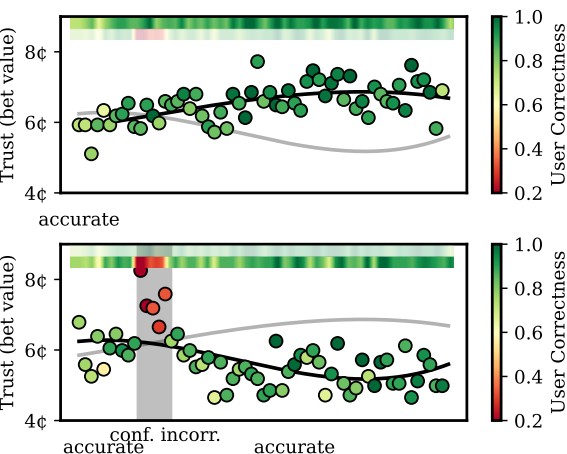

Figure 4: Average user bet values (y-axis) and bet correctness (point & histogram color) with no intervention (control, top) and confidently incorrect intervention (bottom). The spline shows a 3rd degree polynomial fitted with MSE. Transparent features are overlaid from the other graph. See Figure 14 for an annotated version.

We assign each user to a particular condition. For the control group, we show 60 calibrated stimuli. For confidently incorrect and unconfidently correct groups, we show 10 calibrated, then 5 miscalibrated (according to the particular mode), and then 45 calibrated stimuli. We then observe, in particular, the user bet value and accuracy (Figure 4).

**Confidently incorrect intervention.** The control group, which was shown only calibrated stimuli, quickly learns to bet higher than at the beginning and becomes progressively better at it. The confidently incorrect intervention group has the same start but then is faced with the intervention, where they bet incorrectly because of the inaccurate confidence estimation. Even after the intervention, their bet values remain significantly lower and they are worse at judging when the AI is correct. The difference in bet values before and after intervention across confidence levels is also observable in Figure 11. We use the user bet value as a proxy for trust ($\bar{u}^B_{\text{control}} = 7¢, \bar{u}^B_{\text{CI}} = 5¢$) and the user correctness of the bet ($\bar{u}^B_{\text{control}} = 89\%, \bar{u}^B_{\text{CI}} = 78\%$). The significances are $p < 10^{-4}$ and $p = 0.03$, respec-

tively, with two-sided t-test.

Owing to possible errors due to user randomization, we also performed a quasi-experimental analysis of our data to better quantify the effect of our intervention. Interrupted Time Series (Ferron and Rendina-Gobioff, 2014, ITS) analysis is a quasi-experimental method that allows us to assess and quantify the causal effect of our intervention on a per-user basis. ITS models the user's behavior before and after the intervention and quantifies the effect of the intervention. As the comparison is intra-user, it helps mitigate randomness arising from the inter-user comparison between treatment and control. We use ITS with ARIMA modeling, which is expressed as

$$u_t^B = \beta_0 + \beta_1 t + \beta_2 \mathbb{1}_{t>15} + \epsilon_t + \dots$$

where $\mathbb{1}_{t>15}$ is the indicator variable indicating whether $t$ is after the intervention.[2] We are interested in the $\beta_2$ values that indicate the coefficient of deviation from the user bet values before the intervention. Using ITS we find a $\beta_2 = -1.4$ ($p<0.05$ with two-sided t-test), showing a significant drop in user bet value after the confidently incorrect intervention. We thus reject the null hypothesis and empirically verify **H1**.

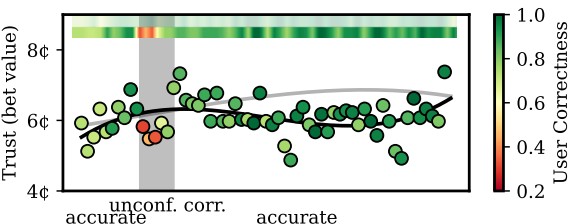

Figure 5: Average user bet values (y-axis) and bet correctness (point & histogram color) with unconfidently correct intervention. The spline shows 3rd degree polynomial fitted with MSE. Transparent features are overlaid from control group (Figure 4, top).

**Unconfidently correct intervention.** We now turn to the unconfidently correct intervention. From Figure 2, this type of intervention is symmetric to confidently incorrect apart from the fact that the baseline model accuracy is 70%. Figure 5 shows that users are much less affected by this type of miscalibration. A one-sided t-test shows a statistically significant difference between the average bet values across control and unconfidently correct

groups ($p<10^{-3}$ with two-sided t-test), which provides evidence for **H2**. Prior work in understanding psychology has found similar results where humans tend to be more sympathetic to underconfident subjects (Thoma, 2016). While applying findings from human-human interaction to human-AI interactions, we exercise caution and acknowledge the need for further research.

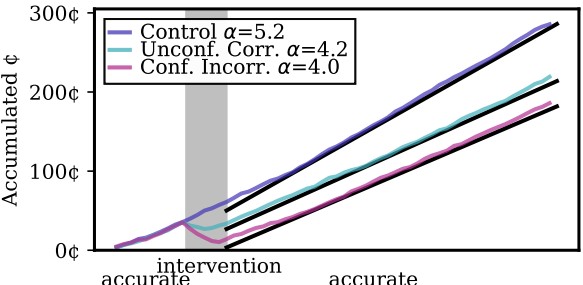

Figure 6: Average accumulated reward. The $\alpha$ is the primary coefficient of linear fits after the 15th stimulus (after intervention). Lines in black are fit using ordinary least squares ($p<10^{-4}$ with two-sided t-test).

**Consequences of lower trust.** We now examine how user's fallen trust in the system affects their task performance. We assert that when the human's trust is calibrated, i.e., the human can effectively decide when the AI is likely to be right or wrong, signifies a strong collaboration. The overall monetary gain, which the user accumulates, acts as a good proxy for the collaboration. To analyze this difference we fit a linear model after the intervention to predict the rate of score increase. We model the cumulative gain at timestep $t$ as $t \cdot \alpha + c$ where $\alpha$ is perceived as the expected gain in ¢ per one interaction. We report $\alpha$ for all three interventions. The results in Figure 6 show that without intervention, $\alpha = 5.2$, which is much higher than with unconfidently correct intervention ($\alpha = 4.2$) and confidently incorrect intervention ($\alpha = 4.0$). Notably, the confidently incorrect intervention has a more negative effect than the unconfidently correct intervention. We thus empirically validate **H3**, miscalibrated examples significantly reduce the performance of the human-AI team in the long run.

> **RQ1 Takeaways:**
> - User trust in the AI system is affected by miscalibrated examples.
> - Confidently incorrect stimuli reduce trust more than unconfidently correct stimuli.

---

[2]We ignore the moving average and error terms for brevity. See Appendix C for the full formula.

| Int. | ¢ | $\alpha$ | $\beta_2$ | $\leq 40$ | | $> 40$ | |
|---|---|---|---|---|---|---|---|
| | | | | Bet | Acc. | Bet | Acc. |
| 0 | 207 | 5.3 | - | 6.6 | 92% | 6.8 | 92% |
| 1 | 188 | 4.8 | $-0.5^\dagger$ | 6.2 | 87% | 6.4 | 88% |
| 3 | 193 | 5.0 | -0.8 | 5.9 | 84% | 5.9 | 82% |
| 5 | 158 | 4.0 | -1.4 | 5.4 | 86% | 5.3 | 90% |
| 7 | 147 | 3.7 | -1.2 | 5.5 | 80% | 5.5 | 86% |
| 9 | 118 | 2.9 | -0.9 | 5.6 | 72% | 5.8 | 84% |

Table 1: Experiments with varying numbers of confidently incorrect stimuli. The $\alpha$ and the gain ¢ are shown from 19th sample (after intervention for all). The columns $\leq 40$ and $> 40$ signify which stimuli in the sequence are considered. All $\beta$ are with $p < 10^{-3}$ with two-sided t-test apart from † which is $p = 0.24$.

## 4.2 Intervention Size

Seeing a noticeable drop in user trust when faced with model confidence errors, we ask:

> **RQ2:** How many miscalibrated examples does it take to break the user's trust in the system?

We do so by changing the number of confidently incorrect stimuli from original 5 to 1, 3, 7, and 9 and measure how much are users able to earn *after* the intervention, how much they are betting immediately after the intervention and later one. We now discuss the average results in Table 1.

Upon observing an increase in intervention size, we note an initial decreasing trend followed by a plateau in $\beta_2$ (4th column), implying a decrease in trust and user bet values, albeit only up to a certain level. Shifting our focus to accuracy, which measures the users' ability to determine the AI's correctness, we observe an initial decline as well (6th column). This decline suggests that users adapt to the presence of miscalibrated examples. However, after 40 examples (25 after intervention), the accuracy begins to rise (8th column) once again, indicating that users adapt once more. Next, we analyze ¢ and $\alpha$, which represent the total reward and the rate of reward increase. As the intervention size increases, both ¢ and $\alpha$ (2nd and 3rd columns) continue to decline. This means that the performance is what is primarily negatively affected. Based on these findings, we conclude that users possess the ability to adapt their mental models as they encounter more calibrated stimuli. However, the decreased trust still leads them to place fewer bets on the system's predictions, resulting in a diminished performance of the human-AI team.

> **RQ2 Takeaways:**
> - even 5 inaccurate confidence estimation examples are enough to long-term affect users' trust
> - with more inaccurate confidence estimation examples, users are more cautious

## 4.3 Mistrust Transferability

Increasingly, single machine learning models are used on a bevy of different topics and tasks (Kaiser et al., 2017; OpenAI, 2023). Owing to the distribution of the training data, the AI's performance will vary over input types. Although users are generally not privy to training data input types, Mozannar et al. (2022) show that users use this variance in model behavior to learn when the model is likely to be wrong. Inspired by this we ask:

> **RQ3:** Do miscalibrated questions of one type of question affect user trust in the model's output for a different type of question?

In the next experiment, we simulate this by having two types of questions – either related to trivia or math. Then, we introduce a confidently incorrect intervention only for one of the types and observe the change in trust on the other one. For example, we introduce a confidently incorrect math questions and then observe how it affects trust on trivia stimuli. We refer to the type of questions we provide intervention for as "affected" questions while the other as "unaffected" questions. We run two sets of experiments where we mix trivia and math as affected questions.

The results in Figure 7 show that there is a gap between trust in the unaffected and affected stimuli type. The gap ($\bar{u}^B_{\text{unaffected}} = 5.4$¢, $\bar{u}^B_{\text{affected}} = 5.0$¢) is smaller than in the control settings (Figure 4) but still statistically significant ($p < 10^{-3}$ with two-sided t-test). This is supported by the analysis using ITS where we look for the relative change to compare user bet values before and after the intervention. We find a significant decrease in bet values for both affected and unaffected questions ($\beta_{\text{affected}} = -0.94$, $\beta_{\text{unaffected}} = -0.53$, $p < 0.05$ with two-sided t-test).

> **RQ3 Takeaways:**
> - Miscalibrated responses of one type affect the user's overall trust in the system
> - Miscalibrated responses of one type further reduce user trust in examples of the same type
> - Thus users also take into consideration question types as they create mental models of the AI system correctness

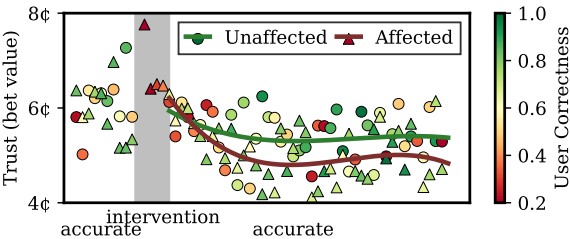

Figure 7: Average user bet values (y-axis) and bet correctness (point & histogram color). The spline shows a 3$^{rd}$ degree polynomial fitted with MSE. 'Affected' is the question type that undergoes confidently incorrect intervention.

| Model | Will agree? | Bet value |
|---|---|---|
| Constant Baseline | 81.8% (69.2%) | 3.2¢ |
| Random Forest (stateless) | 86.8% (81.8%) | 2.9¢ |
| Logistic/Lin. Regression | 87.8% (82.0%) | 2.1¢ |
| Random Forest | 87.9% (82.8%) | 2.0¢ |
| Multi-Layer Perceptron | 87.7% (82.9%) | 1.9¢ |
| GRU | 89.7% (85.0%) | 1.8¢ |

Table 2: Performance of modeling various aspects of user decisions. 'MAE Bet value' column shows mean absolute error 'Will agree?' is formatted as 'F1 (ACC)'. See Section 5 for a description of target variables. 'Stateless' uses only confidence as an input feature.

## 5 Modeling User Trust

In human-AI collaboration systems, it is the collaboration performance that is more important than the accuracy of the AI system itself (Bansal et al., 2021a). In such cases, an AI system that can understand and adapt to how its output is used is more better. An important challenge in understanding the user's behavior is estimating how likely the user is to trust the system. This would also allow the system to adapt when user trust in the system is low by perhaps performing a positive intervention that increases user trust. We apply our learnings from the previous section and show that systems that explicitly model the user's past interactions with the system are able to better predict and estimate the user's trust in the system. We now develop increasingly complex predictive statistical models of user behavior, which will reveal what contributes to the user process and affects trust. For evaluation, we use $F_1$ and accuracy (agreement) and mean *absolute* error (bet value) for interpretability.

- $u_t^D \in \{T, F\}$ Will the user agree? ($F_1$)
- $u_t^B \in [0, 10]$ How much will the user bet? (MAE)

### 5.1 Local Decision Modeling

We start by modeling the user decision at a particular timestep without explicit access to the history and based only on the pre-selected features that represent the current stimuli and the aggregated user history. These are:

- Average previous bet value
- Average previous TP/FP/TN/FN decision. For example, FP means that the user decided the AI system was correct which was not the case.
- AI system confidence
- Stimulus number in user queue

Each sample (input) is turned into a vector,[3] and we treat this as a supervised machine learning task for which we employ linear/logistic regression, decision trees, and multilayer perceptron (see code for details). We evaluate the model on a dev set which is composed of 20% of users[4] which do not appear in the training data and present the results in Table 2. It is important to consider the uninformed baseline because of the class imbalance. The results show, that non-linear and autoregressive models predict the user decisions better although not flawlessly.

Decision trees provide both the importance of each feature and also an explainable decision procedure for predicting the user bet (see Figure 15). They also offer insights into feature importance via Gini index (Gini, 1912). For our task of predicting bet value, it is: previous average user bet (63%), AI system confidence (31%), stimulus number (1%), and then the rest. The $R^2$ feature values of linear regression reveal similar importance: previous average user bet (0.84), AI system confidence (0.78), previous average TP (0.70) and then the rest. The mean absolute error for bet value prediction of random forest models based only on the current confidence (stateless, i.e. no history information) is 2.9¢. This is in contrast to a mean absolute error of 2.0¢ for a full random forest model. This shows that the interaction history is key in predicting user trust.

### 5.2 Diachronic Modeling

Recurrent networks can selectively choose to remember instances of the context that are crucial to making a prediction. Unlike alternate approaches

---

[3]For example, ⟨avg. bet: 6.7, TP: 50%, FP: 10%, TN: 30%, FN: 10%, conf: 81%, i: 13⟩

[4]$(30 + 30 + 30) \cdot 20\% \cdot 60 = 1080$ samples

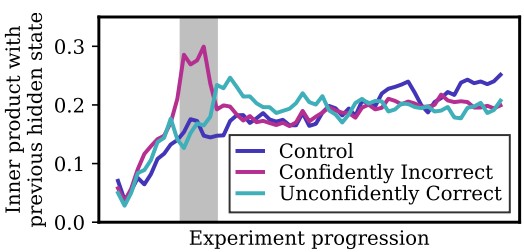

Figure 8: Vector similarity (inner product) between subsequent hidden states of the recurrent GRU model. See Figure 12 for comparison across queues.

that use an average of mean interactions a user had with a system, a GRU can effectively track where user trust in the system underwent a large change. To test this, we look at the information in the hidden state of the GRU we train on the user interactions (see Figures 8 and 12). The GRU's internal state is able to identify areas that caused shifts in the user's trust and changed their future interactions. This peak is much higher for the confidently incorrect than for the unconfidently correct interventions which is in line with our conclusion that confidently incorrect examples deteriorate trust more than unconfidently correct examples.

## 6 Discussion

We now contextualize our findings to real-world applications and discuss the differences and their implications.

**Miscalibration impacts user trust.** Even a small (5) number of miscalibrated examples affects how users trust the system in the future. In our controlled setting we consider a symmetric risk-reward setup. However, past work has shown that trust is linked to risk. In real applications, the reward and cost of trusting the system might not be the same. For example in an AI system detecting cancer, having a doctor manualy do the screening has lower cost than a misdiagnosis.

**Confidently incorrect examples lower trust more than confidently correct examples.** Standard methods of evaluating model calibration, such as the Expected Calibration Error (ECE), do not take this into account. A holistic calibration metric should take these user-centric aspects into account, particularly, how users interpret these confidence scores and how it affects their trust in the system.

**Miscalibration effects persist and affect user behavior over long time spans.** In our setup, users interact with the system continuously over a ses-

sion. After the intervention, their trust decreases over several interactions. Real-life user interactions with AI systems might not always follow this pattern. For example, a user might use a search engine in bursts when they have an information need. The larger time intervals between interactions might dampen the strong feelings of trust or mistrust.

**Mistrust transfers between input types.** Our experiments reveal that the model's miscalibration on a certain type of input also reduces the user's trust in the model on other types of inputs. In real-world applications, AI systems are generally presented to users as an abstraction and the user may or may not be aware of the underlying workings of the system. For example, recent user-facing LLMs often employ techniques such as a mixture-of-experts or smaller specialized models that perform different tasks. In such cases, the transfer of miscalibration can be erroneous.

**RNN outperforms linear models in modeling user trust.** This is indicative that modeling user trust is complex and requires more sophisticated non-linear models. Like most deep learning models, a recurrent network requires more data for accurate prediction. However, user-facing applications can collect several features and with more data deep learning models might generalize better and help us dynamically track and predict user trust.

## 7 Conclusion

When interacting with AI systems, users create mental models of the AI's prediction and identify regions of the system's output they can trust. Our research highlights the impact of miscalibrations, especially in confidently incorrect predictions, which leads to a notable decline in user trust in the AI system. This loss of trust persists over multiple interactions, even with just a small number of miscalibrations (as few as five), affecting how users trust the system in the future. The lower trust in the system then hinders the effectiveness of human-AI collaboration. Our experiments also show that user mental models adapt to consider different input types. When the system is miscalibrated for a specific input type, user trust is reduced for that type of input. Finally, our examination of various trust modeling approaches reveals that models capable of effectively capturing past interactions, like recurrent networks, provide better predictions of user trust over multiple interactions.

## 8 Future work

**Regaining trust.** We examined how miscalibrated examples shatter user trust and we show that this effect persists. We also show that this lack of trust adversely affects human-AI collaboration. Understanding how to build user trust in systems could greatly aid system designers.

**Complex reward structures.** In our experiments, the user is rewarded and penalized equally when they are correct and incorrect. This reward/penalty is also instantly provided to the user. This might not hold for other tasks, for example, in a radiology setting, a false negative (i.e. missing a tumor) has a very large penalty. Past work in psychology has shown that humans suffer from loss-aversion (Tversky and Kahneman, 1992) and are prone to making irrational decisions under risk. (Slovic, 2010). We leave experimentation involving task-specific reward frameworks to future work.

## Ethics Statement

The participants were informed that their data (anonymized apart from interactions) would be published for research purposes and had an option to raise concerns after the experiment via online chat. The participants were paid together with bonuses, on average, $\simeq$\$24 per hour, which is above the Prolific's minimum of \$12 per hour. The total cost of the experiment was $\simeq$\$1500.

**Broader impact.** As AI systems get more ubiquitous, user trust calibration is increasingly crucial. In human-AI collaboration, it is important that the user's trust in the system remains faithful to the system's capabilities. Over-reliance on faulty AI can be harmful and caution should be exercised during deployment of critical systems.

## Limitations

**Simulated setup.** Our experiments were conducted on users who were aware that their actions were being observed, which in turn affects their behavior (McGrath, 1995). We hope our work inspires large-scale experiments that study how users interact directly with a live system.

**Domain separation.** In the Type-Sensitivity Experiment (Section 4.3) we consider only two question types, trivia and math, and provide the participant with an indicator for the question type. In real-world usage, the user might provide inputs that may not be clearly distinct from each other.

**Monetary reward.** A user interacting with an information system to seek information. In our experiments, we replace this goals with a monetary reward. This misalignment in the motivation also affects the participant behavior (Deci et al., 1999).

## Acknowledgments

We thank Hussein Mozannar and Danish Pruthi for their feedback at various stages of the project. We also thank Shreya Sharma, Abhinav Lalwani, and Niharika Singh for being our initial test subjects for data collection. MS acknowledges support from the Swiss National Science Foundation (Project No. 197155), a Responsible AI grant by the Hasler-stiftung; and an ETH Grant (ETH-19 21-1).

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

| Queue | # Users | # Stimuli |
|---|---|---|
| Control | 39 | 2340 |
| Intervention CI 1 | 27 | 1620 |
| Intervention CI 3 | 39 | 2340 |
| Intervention CI 5 | 30 | 1800 |
| Intervention CI 7 | 30 | 1800 |
| Intervention CI 9 | 30 | 1843 |
| Trivia intervention CI | 31 | 1860 |
| Math intervention CI | 31 | 1860 |
| Intervention UC | 40 | 2400 |
| Total | 297 | 17863 |

Table 3: Size summary of collected and released data.

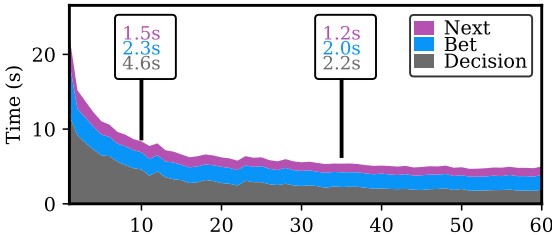

Figure 9: Breakdown of duration of individual user actions. While the bet value decision (bet) and reading the results (next) remain rather constant, the overall decision process becomes faster.

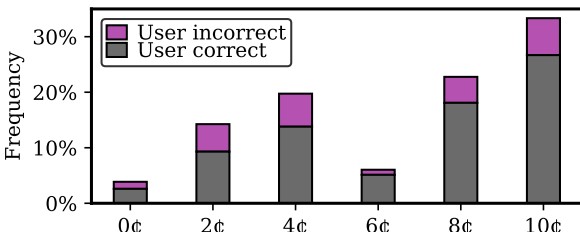

Figure 10: Overall distribution of (bipolar) bet values across all collected data.

## A  Question Generation

We generate 60 trivia and math questions using ChatGPT using the following two prompts. We manually filter questions which are possibly answerable given expert knowledge. See Figure 13 for examples of generated questions. All the generated questions are part of the released data.

*Generate a fake mathematical question that seems like they are answerable but a key information is missing. Generate two plausible definitive answers, the first of which is "correct".*

*Generate fake trivia question that seems like they are answerable but a key information is missing and they are not related to the real world. Generate two plausible definitive answers, the first of which is "correct".*

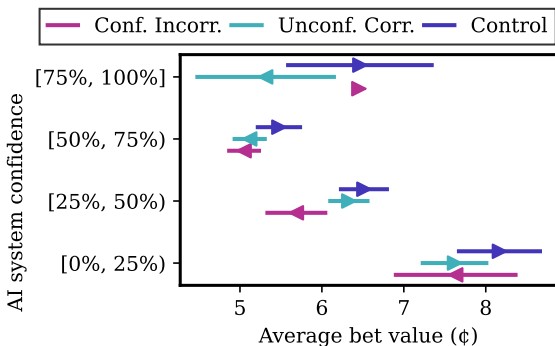

Figure 11: Average bet for a particular confidence interval before and after intervention (▶ means bet increased after intervention and ◀ means decrease). The intervention reduces the bet value which is otherwise naturally increasing. See Figure 10 for bet distribution.

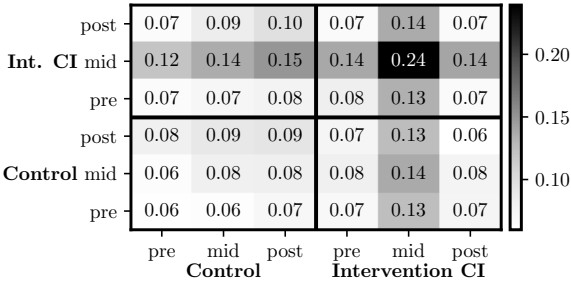

Figure 12: Similarity (inner product) of GRU hidden states for bet value prediction at different locations during the experiment (pre ≤ 10, mid ≤ 15, post > 15) and groups (control/confidently incorrect).

## B  Real AI System Confidence

In all our experiments (control and with different interventions) we find a similar ECE scores (**control**: 0.29% , **UC-5**: 0.30%, **CI-1**: 0.28%, **CI-3**: 0.29%, **CI-5**: 0.28%, **CI-7**: 0.29%, **CI-9**: 0.29%) This is due to the intervention sizes being very small to have a major effect on the ECE score. We compare to other models: DPR on Natural Questions: 37.1%, ResNet-152 on Imagenet: 5.48%.

## C  Interrupted Time Series

$$u_t^B = \beta_0 + \beta_1 \cdot t + \beta_2 \cdot \mathbb{1}(t > 15) +$$
$$\underbrace{\sum_{i=1}^{W} \phi_i u_{t-i}^B}_{\text{Moving average terms}} + \underbrace{\sum_{j=1}^{W} \theta_j \epsilon_{t-j} + \epsilon_t}_{\text{Error terms}}$$

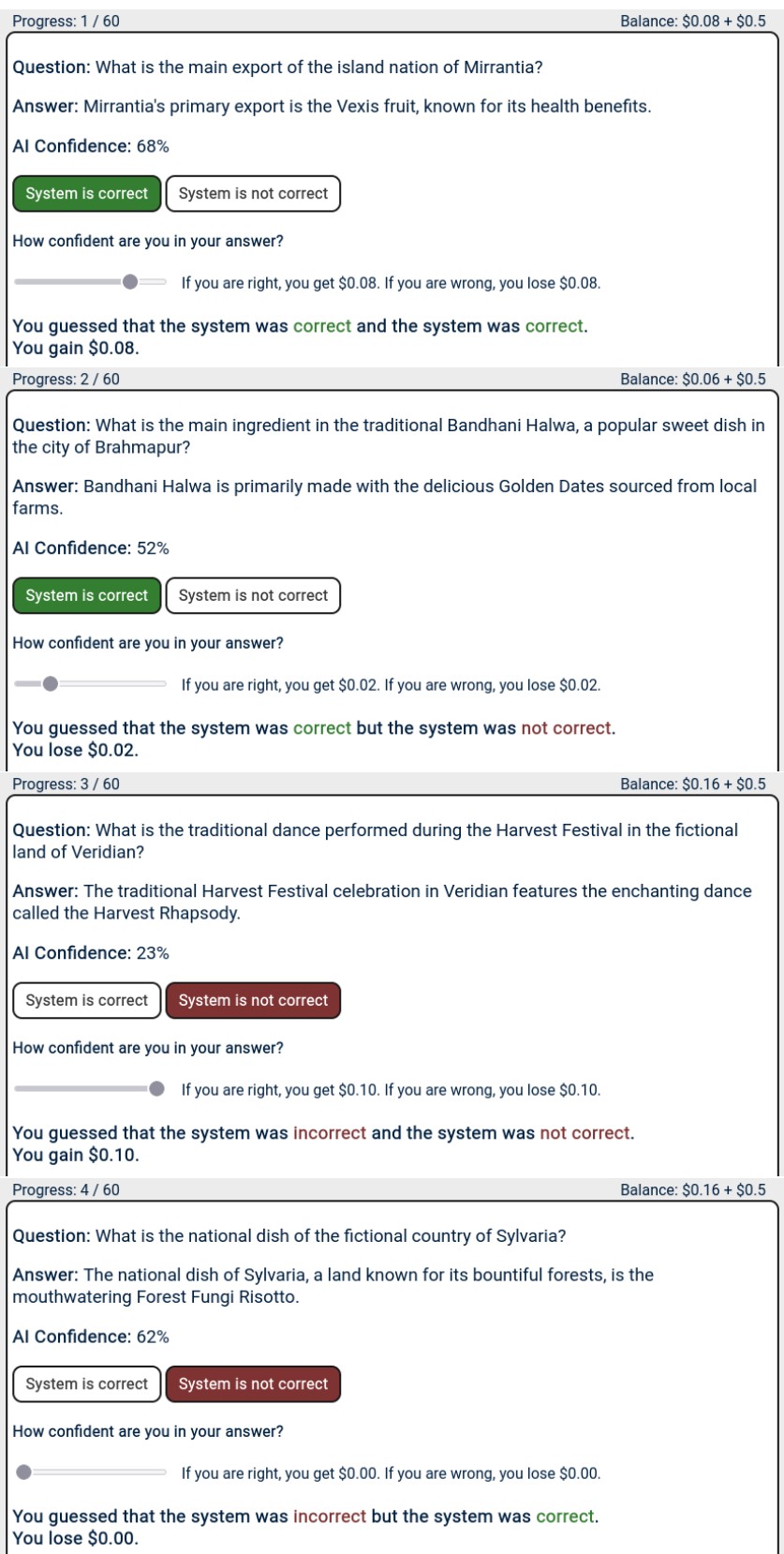

Figure 13: Screenshots of the user interface with all combinations (4) of user guess that the system was correct/incorrect and the model being correct or incorrect.

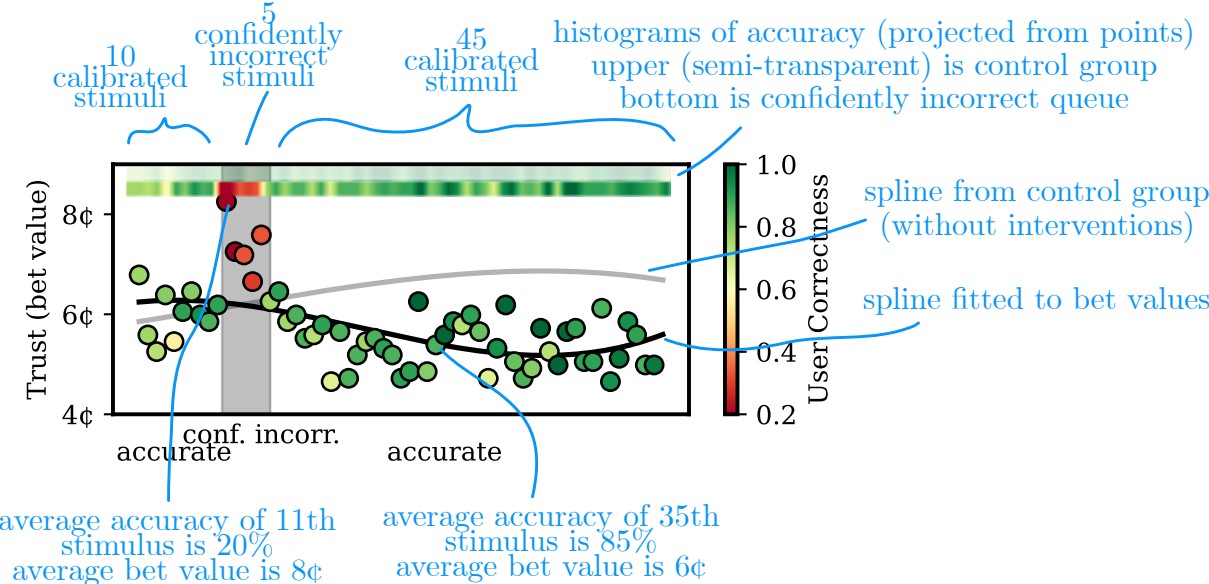

Figure 14: Annotated version of Figure 4. Average user bet values (y-axis) and bet correctness (point & histogram color) with **control** set of stimuli (top) and **confidently incorrect** stimuli (bottom). The spline shows 3rd degree polynomial. Transparent features are overlayed from the other graph.

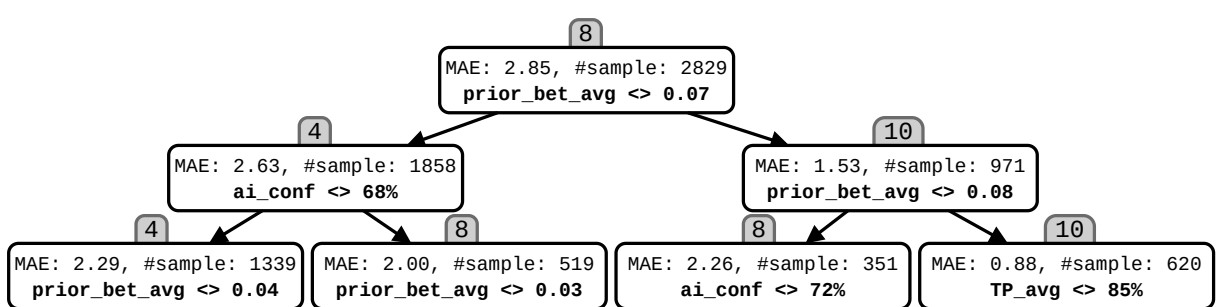

Figure 15: First three layers of a decision tree that predicts bet value (in gray for each node).