# OpenReview forum: "A Diachronic Perspective on User Trust in AI under Uncertainty"
_EMNLP/2023/Conference — EMNLP 2023 Main_

### Official Review · Reviewer_4Q7E · 2023-08-03

**Soundness:** 3

**Excitement:**

2: Mediocre: This paper makes marginal contributions (vs non-contemporaneous work), so I would rather not see it in the conference.

**Missing References:**

- Trust modeling and reputation systems in traditional AI and agent community, e.g., The beta reputation system by Josang.


**Paper Topic And Main Contributions:**

This paper studies how mis-calibrated confidence scores affect user trust in AI systems.  The authors design a monetary betting simulations to study various effects of mis-calibration.  The results show that (1) over-confident wrong responses hurt more than under-confident correct responses, (2) user trust is hard to build but relatively easy to lose, (3) user trust in one question type may be transferrable to the others, and (4) user trust can be modeled.

**Reasons To Accept:**

- The paper is well written, well organized, and easy to follow
- The monetary simulations are promising, new, and interesting


**Reasons To Reject:**

- Most of the findings are expected with limited impact
- Trust modeling with models such as RNN shows good result, but it might be a bit over-kill.  It might be worth comparing traditional simple beta-distribution based computational trust models with confidence estimate in multi-agent system literature.  They are simple, and highly applicable in the settings discussed in this paper, e.g., Evidence-based trust: A mathematical model geared for multi-agent systems by Wang and Singh.

**Reproducibility:**

3: Could reproduce the results with some difficulty. The settings of parameters are underspecified or subjectively determined; the training/evaluation data are not widely available.

**Reviewer Confidence:**

4: Quite sure. I tried to check the important points carefully. It's unlikely, though conceivable, that I missed something that should affect my ratings.

---

> ### Author Rebuttal · Authors · 2023-08-28
>
> Thank you for the review. We are glad that you found our work well written, and our monetary setup to be promising and interesting!
>
> We’ve addressed your concerns below:
>
> **Impact and contributions.**
> We agree that miscalibration’s link to trust is expected. However, several of our follow-up research questions do provide interesting insights that are novel in context of past literature in HCI.
> - The diachronic effects of miscalibration on trust have not been explicitly studied. Several recent papers [1,2, inter alia] only study the effects of confidence scores on trust and neglect the temporal effects.
> - We show that miscalibrations in the past can have detrimental effects on user trust which can reduce the user-AI performance in the future.
> - Our findings on the difference in effect between confidently incorrect and unconfidently correct examples is novel. Current calibration metrics (e.g. ECE, Brier score) treat overconfidence and underconfidence as the same.
> - Our extensions to understanding user trust with miscalibrations in the context of different input types builds on very recent work [3] on how users create mental models over different types of inputs.
>
>
> **Multi agent computational trust.**
> Thank you for suggesting work in trust modeling from the multi-agent system community [4,5]. This line of research has also been missed in several recent HCI works that model user trust. We will update our related work section to include papers about modeling trust from the traditional AI and multi-agent community.
>
> **RNNs vs Beta dist trust model.**
> We ran initial experiments using a simple beta distribution trust model (BDTM) to predict the probability of the user trusting the AI system (Table 2 in our work). We used the description [here](https://arxiv.org/pdf/1806.03916.pdf) to implement the model. We observe an accuracy of 78%. Compared to Table 2 in the paper, the BDTM performs better than a constant baseline (69%) and is in the range of a single layer logistic regression model (82%)  but it underperforms the RNN (85%).
> One of our major conclusions from the trust modeling experiments is that an RNN  given its ability to be stateful can better model the user’s trust which we show in the prior sections can be drastically changed by a small intervention.
> In our experiments we show that the RNN outperforms models that only have access to the averages of the past.
>
> Although BDTM's performance could be enhanced with more intricate techniques for weighting and discounting past experiences, in our context with just 2 agents, a recurrent method like the RNN gives us a powerful and flexible function that can be easily trained from past experiences. We will add a more detailed discussion regarding this to the paper.
>
>
> - [1] Vodrahalli, Kailas, Tobias Gerstenberg, and James Y. Zou. [Uncalibrated models can improve human-ai collaboration.](https://proceedings.neurips.cc/paper_files/paper/2022/file/1968ea7d985aa377e3a610b05fc79be0-Paper-Conference.pdf) Advances in Neural Information Processing Systems 35 (2022): 4004-4016.
> - [2] Gonzalez, Ana Valeria, et al. [Human evaluation of spoken vs. visual explanations for open-domain qa.](https://arxiv.org/abs/2012.15075) Association of Computational Linguistics ACL (2021).
> - [3] Mozannar, Hussein, Arvind Satyanarayan, and David Sontag. [Teaching humans when to defer to a classifier via exemplars.](https://arxiv.org/pdf/2111.11297.pdf) Proceedings of the AAAI Conference on Artificial Intelligence. Vol. 36. No. 5. 2022.
> - [4] Josang, Audun, and Roslan Ismail. [The beta reputation system.](https://domino.fov.um.si/proceedings.nsf/0/d9e48b66f32a7dffc1256e9f00355b37/$FILE/josang.pdf) Proceedings of the 15th bled electronic commerce conference. Vol. 5. 2002.
> - [5] Wang, Yonghong, and Munindar P. Singh. [Evidence-based trust: A mathematical model geared for multiagent systems.](https://dl.acm.org/doi/10.1145/1867713.1867715) ACM Transactions on Autonomous and Adaptive Systems (TAAS) 5.4 (2010): 1-28.

---

### Official Review · Reviewer_3PXu · 2023-08-04

**Soundness:** 3

**Excitement:**

3: Ambivalent: It has merits (e.g., it reports state-of-the-art results, the idea is nice), but there are key weaknesses (e.g., it describes incremental work), and it can significantly benefit from another round of revision. However, I won't object to accepting it if my co-reviewers champion it.

**Paper Topic And Main Contributions:**

This paper studies users' mental model of the AI system in a human-AI collaboration enviroment. The paper is motivated by the negative impact of incorrect prediction on user trust and the need of understanding the dynamics of this process. The authors conduct a user study where users interact with a question answering system. Users' trust is proxied using monetary rewards. The paper uses a simulated AI to control question generation, confidence, and answer correctness. The authors conduct a set of tests to verify their hypothesis. Based on the findings, the authors create a classifier and a regression model to predict users' agreement and bet. The authors also use a GRU model to capture the impact of interaction history.

**Reasons To Accept:**

The paper is well-organized and easy to follow.
The paper is well motivated and the problem of modeling user trust is important and challenging for applying AI models.
The paper conducts a good amount of experiments with interesting research questions.

**Reasons To Reject:**

The user study is running on a simplified setting, e.g., users do not know the answers, users do not have any prior bias. Given such an impractical setting, the contribution of the findings might be limited.

**Reproducibility:**

4: Could mostly reproduce the results, but there may be some variation because of sample variance or minor variations in their interpretation of the protocol or method.

**Reviewer Confidence:**

3: Pretty sure, but there's a chance I missed something. Although I have a good feel for this area in general, I did not carefully check the paper's details, e.g., the math, experimental design, or novelty.

---

> ### Author Rebuttal · Authors · 2023-08-28
>
> Thank you for your review. We concur that studying user trust is an important area. We are also happy that you found the paper well organized, our work well motivated, and our experimentation thorough.
>
> We have addressed your concerns below:
>
> **Excluding prior bias.**
> As reviewer _Xqzv_ points out, we made deliberate efforts to control for prior knowledge of the participants in order to isolate the effect of the model's reported confidence on user trust and productivity. Without this, we would be introducing a confounder which will make it challenging to answer our research questions. We acknowledge that in the real world the users may have some partial knowledge regarding the answer which they could use in their decision-making.
>
> We believe that our work enables future studies which attempt to control for the effect of prior knowledge and bias. For example, deliberately providing answers which _conflict_ with the user’s prior knowledge and measure this effect in comparison to trust loss caused by confidently incorrect or unconfidently correct outputs.

---

### Official Review · Reviewer_Xqzv · 2023-08-12

**Soundness:** 4

**Excitement:**

5: Transformative: This paper is likely to change its subfield or computational linguistics broadly. It should be considered for a best paper award. This paper changes the current understanding of some phenomenon, shows a widely held practice to be erroneous in someway, enables a promising direction of research for a (broad or narrow) topic, or creates an exciting new technique.

**Paper Topic And Main Contributions:**

In this paper the authors investigate how humans trust AI agents over time with increasing number of engagements. The experiment design involves human participants  interacting with agents in multiple rounds of question answering. In each round a participant is presented with a question together with the model's answer and output confidence. Given this information, the task is to either accept or reject the output. Each time the particpant is also asked to place a monetary bet on the correctness of the model's answer, serving as a proxy for trust. In this controlled experiment, the authors use ChatGPT to generate fictional questions not grounded in reality and also control model calibration.

As a major contribution of this work, the authors show that after participants are presented with miscalibrated output (unconfidently correct, UC; or confidently incorrect, CI) the human trust in the agent decreases, which is reflected by a decrease in bets. Other contributions include (1) showing that CI harms user trust more than UC, (2) investigating the interaction between the frequency of subsequent bad interactions, i.e. miscalibrated outputs, and user trust and, (3) the impact of miscalibrated outputs on the overall trust across different question types. Lastly, the authors propose a way to model user trust over time.


**Reasons To Accept:**

1. I truly enjoyed reading this paper, it's extremely well written, well structured and presents a very interesting experimental design.
2. This paper is likely to inspire a lot of follow up work. Trustful LLMs is a growing area and the experiments presented in this paper could easily extended to other tasks, (synthetic/real) datasets and user groups, as pointed out in §8. As such, I believe this paper is impactful to a large part of the ACL community.
3. The authors control for confounding factors such as real world prior knowledge, monetary reward instead of asking participants' confidence in the answer, thereby establishing a clean test environment sufficient to support the presented hypotheses.


**Reasons To Reject:**

1. As pointed out by the authors (L616-629), the experimental setup presents a low-stake scenario, the trust function (approximated by a monetary utility function) would look different in medium and high-stake scenarios.
2. User trust is also impacted by factors other than self-reported model confidence, model output and the information about the correctness of the answer, including, e.g., the nature of the task and the model's ability to explain the output generation process.
3. The study assumes access to an oracle that informs users if the model answer is correct ("AI correctness revealed"), impacting the mental model. This assumption often does not hold in real world scenarios. If we had access to an oracle, we would not need to interact with the agent. If we don't have access to an oracle, we could not update the mental model by revealing the answer.


**Reproducibility:**

4: Could mostly reproduce the results, but there may be some variation because of sample variance or minor variations in their interpretation of the protocol or method.

**Reviewer Confidence:**

4: Quite sure. I tried to check the important points carefully. It's unlikely, though conceivable, that I missed something that should affect my ratings.

---

> ### Author Rebuttal · Authors · 2023-08-28
>
> Thank you for the very favorable review. We are glad you enjoyed reading our work and found it impactful!
>
> We have addressed your concerns below:
>
> **Medium and high stakes scenarios.**
> We agree that a current limitation of this study is that it only studies cases where the risk-reward structure is symmetric.
> We conducted a very preliminary study where we provided asymmetric rewards (i.e. higher penalty for being incorrect than reward for being correct and vice versa). For confidently incorrect intervention size of 5 we observe a higher mean bet value (0.72 vs 0.63) when we double the reward for the same risk, i.e., users are willing to bet more as the possible reward is higher.
> When increasing the penalty we however do not notice a commensurate decrease in bet values. These results demonstrate a need for future research into how the risk to reward ratio can affect user interactions with AI systems. We are happy to incorporate a discussion on this.
>
> **Other Factors affecting user trust.**
> We fully agree that model explanations are one of the key contributors to establishing user trust (L096-L100). However, we consider it to be beyond the scope of our current work and leave studying the diachronic effects of model explanations on user trust as future work.
>
> **Access to oracle answers.**
> We acknowledge that having an oracle give you immediate feedback is not realistic. However we believe that there still is a feedback loop (i.e. the user is eventually informed if the AI was correct or not) albeit not immediately. Using an oracle to reveal the answer (immediate feedback) is commonly used in literature [1,2, inter alia]. Using a delayed feedback would introduce other sources of noise and we leave these investigations, which would explicitly control for this factor, to future work.
>
> - [1] Gagan Bansal, Tongshuang Wu, Joyce Zhou, Raymond Fok, Besmira Nushi, Ece Kamar, Marco Tulio Ribeiro, and Daniel Weld. 2021. [Does the Whole Exceed its Parts? The Effect of AI Explanations on Complementary Team Performance](https://dl.acm.org/doi/pdf/10.1145/3411764.3445717). In Proceedings of the 2021 CHI Conference on Human Factors in Computing Systems (CHI '21). Association for Computing Machinery
> - [2] Gonzalez, Ana Valeria, et al. [Human evaluation of spoken vs. visual explanations for open-domain qa.](https://arxiv.org/abs/2012.15075) Association of Computational Linguistics ACL (2021).

---

### Meta-Review · Area_Chair_SXih · 2023-09-13

**Recommendation:** 4

**Metareview:**

This paper presents a well-thought-out user study about how model miscalibration affect the user's trust over multiple trials. All reviewers agree that the paper is well-written, the research question is important, and that the experimental setup of using bets to measure trust is novel. Reviewer 4Q7E recommended beta-distribution based computational trust models as baselines, which is addressed by the authors. The only downside of this paper is the generalizability of results from the simplistic setting, but the authors made reasonable efforts to improve that by controlling for the user's prior knowledge. This paper presents a good blueprint for future work studying the temporal effect of calibration on trust in AI.

For related work, I recommend including HCI literature on the impact of feedback on user trust, e.g., No Explainability without Accountability: An Empirical Study of Explanations and Feedback in Interactive ML by Smith-Renner et.al.

---

### Decision · Program_Chairs · 2023-10-07

**Decision:**

Accept-Main

**Comment:**

This paper presents a well-thought-out user study about how model miscalibration affect the user's trust over multiple trials. All reviewers agree that the paper is well-written, the research question is important, and that the experimental setup of using bets to measure trust is novel. Reviewer 4Q7E recommended beta-distribution based computational trust models as baselines, which is addressed by the authors. The only downside of this paper is the generalizability of results from the simplistic setting, but the authors made reasonable efforts to improve that by controlling for the user's prior knowledge. This paper presents a good blueprint for future work studying the temporal effect of calibration on trust in AI.

For related work, I recommend including HCI literature on the impact of feedback on user trust, e.g., No Explainability without Accountability: An Empirical Study of Explanations and Feedback in Interactive ML by Smith-Renner et.al.